# PROCEEDINGS A

microsystems, nanotechnology, biomedical engineering

COVID-19, SARS-CoV-2, microfluidic devices, microfabrication, nanofabrication, viruses

**Author for correspondence:**
José Alvim Berkenbrock
e-mail: j.alvim@usask.ca

# Microfluidic devices for the detection of viruses: aspects of emergency fabrication during the COVID-19 pandemic and other outbreaks

José Alvim Berkenbrock[1], Rafaela Grecco-Machado[2] and Sven Achenbach[1]

[1]Department of Electrical and Computer Engineering, and
[2]Department of Anatomy, Physiology and Pharmacology, University of Saskatchewan, Saskatoon, SK, Canada

JAB, 0000-0001-5567-4449

Extensive testing of populations against COVID-19 has been suggested as a game-changer quest to control the spread of this contagious disease and to avoid further disruption in our social, healthcare and economical systems. Nonetheless, testing millions of people for a new virus brings about quite a few challenges. The development of effective tests for the new coronavirus has become a worldwide task that relies on recent discoveries and lessons learned from past outbreaks. In this work, we review the most recent publications on microfluidics devices for the detection of viruses. The topics of discussion include different detection approaches, methods of signalling and fabrication techniques. Besides the miniaturization of traditional benchtop detection assays, approaches such as electrochemical analyses, field-effect transistors and resistive pulse sensors are considered. For emergency fabrication of quick test kits, the local capabilities must be evaluated, and the joint work of universities, industries, and governments seems to be an unequivocal necessity.

## 1. Introduction

The preparation of this manuscript started amid one of the most severe virus outbreaks of our recent

history. Since the identification of the new coronavirus (SARS-CoV-2) reported in December 2019, it rapidly spread worldwide [1]. Between 30 January and 11 March, the World Health Organization (WHO) changed the COVID-19 outbreak status from 'Public Health Emergency of International Concern' to 'pandemic' [2,3]. In a couple of months, the epicentre of this disease moved from Wuhan in the Province of Hubei, China, to Lombardy in Italy with about 800 deaths/day and a total of 10 000 fatalities as of 27 March 2020 [4]. A few days later, the USA became the epicentre of the new coronavirus when around 103 000 cases and 1668 deaths were reported at the time [4]. By 12 September 2020, there were more 6 300 000 cases, and about 191 000 deaths reported in the USA related to this disease [5]. In a different direction, countries such as Singapore and South Korea did not experience the same trend in fatality rate numbers. The rapid and wide use of screening tests has been considered key to their achievements [6–8]. In these countries, large efforts were made to increase the availability of test kits and to screen the population and isolate the infected citizens. The current good performance of these countries has been attributed to the lessons learned from past outbreaks (e.g. SARS, MERS) [9,10].

Although the wide testing of people has been considered a key action for containing the virus spread, it is also the bottleneck for many countries around the world. There are many countries in Africa and the Americas that even with large populations, have less than 1000 reported cases. The shortage of screening tests reported by some of these countries could be a factor masking up the actual number of cases [8,11,12]. A massive number of at-home screen test kits are urgently needed for individual isolation since few countries are able to apply strict lockdowns [7]. The current standard for confirming presumptive cases of SARS-CoV2, endorsed by the WHO, applies a real-time quantitative reverse transcription-polymerase chain reaction (qRT-PCR) method, which requires primers, thermocyclers, specific reagents and qualified personal [8,13]. Even though these might be considered common items to be found in healthcare institutions, the gap between need and availability of such diagnostics kits worldwide, and the necessity in many countries to import such tests might indicate otherwise.

An alternative to the time-consuming bench assays can be found in the field of microfluidics. Within a few decades, microelectronics and micro-electromechanical systems (MEMS) technologies have enabled the emergence of microfluidic devices capable of manipulating minute amounts of fluids and extracting information from it [14,15]. This approach also offers the potential to rapidly obtain information from the small sample volumes. It has increasingly been used for bedside or point-of-care testing (PoCT). Many microfluidic devices have been developed for early diagnosis of diseases or other health-related conditions (e.g. pregnancy test, glucose levels, pneumonia) by the detection of target elements in the circulation [15,16]. In recent years, microdevices have been developed to also detect smaller pathogens like viruses [17,18]. In this work, we review the state-of-the-art in microfluidics devices for the manipulation and detection of infectious agents such as the coronavirus, the human immunodeficiency virus (HIV), influenza viruses and the Zika virus. We discuss the detection approaches and the signalling methods used for identifying specific targets and informing the user about the detection. Moreover, we analyse similarities to the benchtop assays, and we end this work presenting a perspective on how such devices could be used in emergency fabrication situations box 1.

## 2. Recent outbreaks

In recent history, no other virus caused such a disruption to our social system as the current outbreak of the COVID-19 disease caused by the SARS-CoV-2 virus. In December 2019, this new virus was first identified from human airway epithelial cells of patients in Wuhan, China. The virus (GCF_009858895.2) was recognized as being a positive-sense RNA virus (i.e. a viral RNA that can be immediately translated by the host cell) with about 29 700 base pairs and a non-segmented envelope [1]. Coronaviruses are commonly found in birds and mammals and usually cause only mild symptoms to humans [11]. The new virus is in the same genus as Severe Acute Respiratory Syndrome (SARS-CoV) and Middle East Respiratory Syndrome (MERS-CoV) [11,24]. Both, SARS-CoV and MERS-CoV, led to outbreaks in the last couple of decades.

**Box 1.** The polymerase chain reaction (PCR) method and its variations.

**PCR**: a technique widely used to accurately and rapidly generate a large number of copies (amplification) of a DNA target sequence. The amplified DNA product can be used in a variety of ways, including for the diagnosis of infectious diseases [19]. PCR reactions need a DNA template, primers that bind to the target DNA through complementary base pairing, an enzyme called DNA polymerase which synthesizes new complementary DNA strands, and nucleotides that will compose the new strands. All these reagents are combined and submitted to repeating heating and cooling cycles. Each cycle doubles the amount of DNA, resulting in an exponential DNA increase [20]. The number of cycles can vary between 25 and 40 until achieving a reasonable number of copies.

**Real-time PCR**: also known as quantitative PCR (qPCR), this technique allows for quantification of the target DNA while the reaction is happening. Real-time detection is possible because of the use of fluorescent reporter molecules that bind to double-stranded DNA during the PCR cycles [19]. Levels of fluorescence increase with an increasing amount of PCR product. Fluorescence readings are performed with specialized thermal cyclers and the data needs to be analysed by instrument-specific software.

**Reverse transcription PCR (RT-PCR):** uses the same amplification principle as the traditional PCR, but differs in the sample preparation stage. While the PCR amplifies the fragments of DNA, RT-PCR is used for the amplification of RNA targets. The RNA templates are first converted into complementary DNA (cDNA) and then used as templates in PCR reactions [21]. The amplification of RNA strands is relevant for the study of transcribed gene sequences and RNA-based viruses [5,22,23].

A combination of qPCR and RT-PCR (qRT-PCR) was endorsed by the WHO to detect COVID-19 earlier this year [13].

Many questions remain about the behaviour of SARS-CoV-2, including how its mortality rate compares to other coronaviruses. It is especially difficult to predict the mortality rate of a current outbreak due to the uncertainties inherent to an evolving process. One limitation of the estimations on the mortality rate is assessing the actual number of infected people [8,25]. Current estimations of the mortality rate for COVID-19 vary from 0.9% up to 8% or more [12,25]. For patients in critical care, it has reached levels as high as 52% in Washington (USA) and 62% in Wuhan (China) [26]. Most countries were not able to extensively test the population in the rise of COVID-19, diminishing the accuracy of mortality rate numbers in the wider population. Some places have therefore later employed home-tests to assess the number of people who have already developed antibodies against SARS-CoV-2 [27–29]. This afterward-testing can contribute to an improved approximation of the actual mortality rate. In comparison with other coronaviruses, between 2003 and 2004, SARS caused close to 800 deaths worldwide with a mortality rate lower than 10% [1,11,24,30]. In 2012, MERS emerged causing a similar number of fatalities, however, with a mortality rate higher than 34% [1,11,24].

Along with outbreaks caused by new strains of coronaviruses, emergent novel influenza strains have also imposed threats globally [31]. The number of deaths related to influenza per year is estimated to be between 291 000 and 646 000 with a mortality rate ranging from 4.0 to 8.8 per 100 000 individuals (i.e. 0.004% to 0.009%) [32]. The highest influenza mortality rate is among older patients (51.3 to 99.4 per 100 000 individuals). However, about 9243–105 000 deaths were estimated among children younger than 5 years, too [32]. In the ongoing SARS-CoV-2 pandemic, COVID-19 killed more than 917 000 people worldwide in 9 months and infected about 28 million people.

There are other viral infections considered epidemics that present local and global threats to human health. Dengue is a wide-spread disease that has affected more than 100 countries and has caused 10 000 deaths worldwide per year [33]. Other viral outbreaks have been restricted to some areas of the globe, such as Chikungunya, Zika and Ebola viruses. They caused, however,

numerous fatalities and resurface over the years. The fatality rate of Chikungunya (1/1000) is higher than that of dengue (0.5/1000) and lead to about 200 deaths just in Brazil in 2016 [34,35]. Between 2015 and 2016, an outbreak of Zika was reported in the Pacific Islands and American countries [36,37]. The morbidity caused by Zika can lead to disabilities lasting a lifetime [37]. For Ebola, the fatality rates reported by the WHO varied from 25% to 90% in the past. A total of 28 616 cases have been reported in Guinea, Liberia and Sierra Leone, with 11 310 deaths [38]. Ebola continues to resurface and in 2018, about 50 deaths were attributed to the virus in the Democratic Republic of the Congo [39]. These numbers indicate the importance of creating detectors that can be largely produced and used among populations with different societal, economic and cultural backgrounds.

## 3. Fabrication aspects for microfluidic systems

In this section, some common fabrication techniques for microfluidic devices are reviewed. The discussion focuses on the most frequently used techniques for the fabrication of microfluidic detectors, which include printing and cutting, moulding and photolithography.

### (a) Printing and cutting

Paper-based microfluidic detectors are robust, cost-effective and user-friendly devices [40,41]. They can be used for the diagnosis of a variety of viruses [41–47]. There are many advantages in using paper-based devices: *e.g.* this material is readily available worldwide, and its properties allow for easy transport of liquids using passive flow [41]. Also, various types of paper are compatible with printing and other patterning technologies, increasing their applicability [41–44]. An example of a paper-based device for viral detection was presented by Magro *et al.* [43] for the diagnosis of the Ebola virus. Using wax on folding paper and double-stick tape, the group created a device that allowed a multiplexed analysis (figure 1). The fabrication steps were printing and cutting, adding RT-RPA[1] reagents, and freeze-drying the device. The devices could be stored and delivered to locations in need, although it still required RNA extraction of filtered blood as a first analysis step [43]. Paper-based devices were also used for the diagnosis of mosquito-borne RNA viruses such as Zika, Dengue and Chikungunya [42,44–47]. Batule and co-workers [44] presented a multi-step process for detecting these viruses consisting of two devices. The first one allowed the extraction of viral RNA based on single-strand DNA probes from the sample, while the second device was used to perform an RT-LAMP (loop-mediated isothermal amplification[2]) assay for sample amplification. Although the complete system requires a fluorescence imaging system for reading, the results were comparable to RT-qPCR. Even though much has still to be researched on paper-based immunosensors, these biosensors are cheap, easy to handle, biocompatible and biodegradable, which can make them a promising option to be widely used in regions with limited resources.

### (b) Moulding

Different micro-replication techniques can be used for the fabrication of a large number of devices usually made of polymers [49]. Variations of compression moulding (i.e. embossing) and injection moulding are widely employed for the fabrication of microfluidic devices [5,50–55]. Fernández-Carballo *et al.* [5] fabricated a mould in steel for high-volume manufacture of an RT-PCR chip. Using this mould, the authors suggested that 200 chips could be produced per hour [5]. This ability to produce several devices with a single mould would be advantageous

---

[1]Real-time recombinase polymerase amplification (RT-RPA) is an isothermal method for genetic material amplification performed at temperatures between 25 and 42°C. Introduced in 2006, the RPA is a rapid and cost-efficient method that allows for $10^4$-fold amplification in 10 min. Minimum sample-preparation, commercial availability of freeze-dried reagents and low-temperature operation are some advantages of this method [48].

[2]Loop-mediated isothermal amplification (LAMP) is briefly explained below in the sub-section *État des choses*.

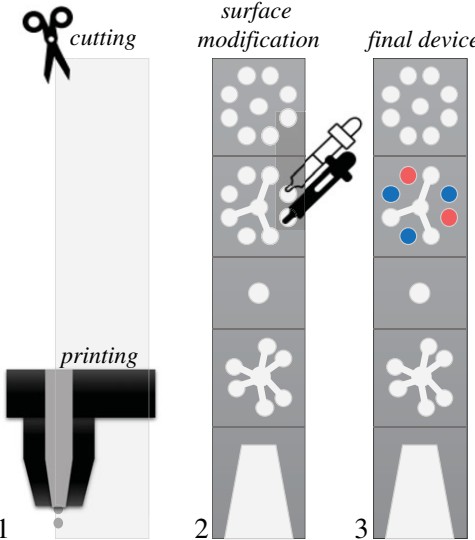

**Figure 1.** Example of fabrication steps for a paper-based device using printing and cutting processes. Adapted from [43]. (Online version in colour.)

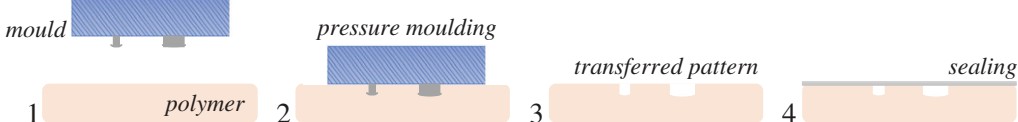

**Figure 2.** Example of a fabrication process of a microfluidic device based on compression moulding. A mould was fabricated using photolithography and is used to imprint a pattern into a polymer substrate. (Online version in colour.)

for localities with difficult access to sophisticated microfabrication laboratories. Another example of fluidic chips fabricated using the moulding process was presented by Wang *et al.* [51]. The team produced a LAMP-integrated chip on polycarbonate with microchannels (500 μm wide and 200 μm deep) and overall size of $2 \times 1 \times 1\,mm^3$ [51]. Chip fabrication using moulding processes for tailoring polymers (figure 2) is simpler than photolithography and the process itself induces lower costs [53]. Some disadvantages of the moulding process are related to the required heat and pressure which may lead to tool deterioration, polymer sample deformation/shrinkage and other fabrication defects [53].

## (c) Photolithography

Photolithography is a very well-established process used in micro- and nanofabrication. The fundamental elements in this process include a light source (e.g. ultraviolet radiation), a mask and a substrate coated with a photoresist (i.e. photosensitive material) [15,49]. In brief, a pattern on the mask is transferred to the photoresist by exposure to the UV radiation. In the resist, a chemical reaction is catalysed with the UV photons (figure 3). After eventual thermal treatment (baking steps), wet chemical development selectively removes either the exposed or the unexposed resist areas (depending on the type of resist). These basic fabrication steps of lithography deliver a primary polymer microstructure. It can be used as fluidic channels and hence, after appropriate sealing, be used as the final microfluidic device. Alternatively, the resist pattern can be applied as a processing mask in subsequent processing steps, such as is etching or electroplating. Additional

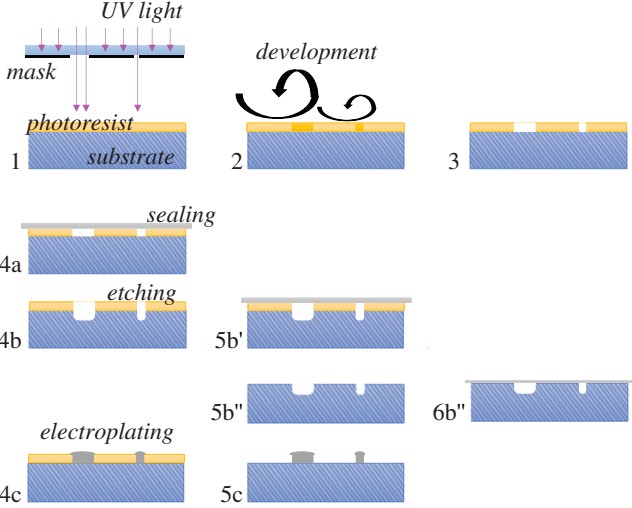

**Figure 3.** Process steps for the fabrication of a microfluidic device using photolithography and subsequent processes. (i) Using exposure to UV radiation, a mask pattern is transferred to a photoresist which was previously coated on a rigid substrate. (ii) The sample is wet chemically developed, i.e. either the exposed or the unexposed resist areas are selectively removed. Depending on the resist, intermediate thermal treatment steps (baking) may be required. This delivers the primary polymer microstructure, which may serve as the final fluidic device (see process alternative 4a), or it may serve as a processing mask (see all other subsequent processing steps below. (3) After these core lithographic steps, different subsequent processing options are available: the patterned substrate can be sealed (4a), or further processed (4b, 4c). In process variation (4b), the substrate is locally etched, using the resist pattern as a processing mask, and the device can be sealed (5b′). If needed, the resist can be removed (stripped) before sealing (5b′′, 6b′′′). In process variation (4c), the resist is used as a processing mask to locally grow metal microstructures by electroplating. This process variation is commonly applied to create a mould which can be used in a moulding process described in the previous paragraph. (Online version in colour.)

fabrication steps required in the processing of a detector system may include the generation of electrodes and eventually functionalization of electrodes for the detection of biomolecules [53,56–58]. An example of these extra processing steps was presented by Iswardy *et al.* [56]. Silica beads were functionalized with antibodies (i.e. 4G2) to bind to the Dengue virus (DENV) [56]. The operation of the device was based on dielectrophoretic forces to capture 4G2-coated silica beads flowing in a microchannel. The beads were mixed with the samples, then captured with V-shaped electrodes and imaged with an inverted fluorescent microscope. In this case, the substrate used underneath the microchannel was glass, a common material in microfluidics. Photolithography is a versatile technique that can be performed with a variety of materials [56,58–62]. It can be part of a larger fabrication process sequence that may encompass etching, lift-off, electrochemical deposition and doping, for instance [53]. Sophisticated technologies and equipment are required for photolithographic processing. Among common fabrication processes for microfluidics, photolithography typically has the strictest cleanliness requirements, since particle contamination may interfere in pattern transfer, and surface contamination may reduce photoresist adhesion to the substrate [53]. Mass production of microdevices can be simplified, and requirements on the fabrication environment can be relaxed, by omitting the continued use of lithographic approaches when moulds are created. In this case, this lithographic process would only be applied once, and with subsequent electroplating (figure 3, variation 4c). The resulting moulds then still feature lithographic accuracy, but allow for simple and fast replication of the predefined pattern, see the description of the moulding processes above [63–66]. However, modifications to the mould design require a new round of photolithography fabrication, which includes creating another photomask and all the other steps described earlier.

Microfabrication technologies have evolved and have been refined and differentiated, to respond to added requirements imposed by microfluidic sensing/detection tasks. Besides using the most common fabrication processes presented above, microfluidic devices have also been created using, e.g. three-dimensional printers [44,67,68], focused ion beam (FIB) patterning [69–71] or controlled breakdown (CBD) of membranes [70,72,73]. Some detector designs require highly sophisticated fabrication technologies, but many effective new microdevices have been envisaged using simple and largely available technologies like three-dimensional printing or regular printing for paper-based devices.

# 4. Detection approaches using microfluidic systems

## (a) Early stages of biological assays on a chip

Viruses are generally formed by their genetic codes (i.e. DNA or RNA) and comprise a protein-made shell and lipid ligands. Detection approaches may be endeavoured for any of these elements. Traditional benchtop assays, such as enzyme-linked immunosorbent assay (ELISA) and western blot are largely used to detect proteins, while polymerase chain reaction (PCR) and its variants are used for the detection and amplification of genetic material. These methods require several types of biotechnological supplies as buffers, genetic-probes, antibodies or fluorescent labels, as well as laboratory plasticware, benchtop equipment and highly trained personnel [74,75]. Despite, or because, of these downsides of a large number of materials and training required, these traditional techniques excel in accuracy, sensitivity and reliability. A variety of approaches based on benchtop assays can be adapted to microfluidic devices to reduce the quantity of required reagents and to speed up reactions. The translation of traditional techniques into miniaturized versions seemed to be an almost natural developmental step since they apply well-established protocols. In the mid-1990s, the feasibility of a PCR on-a-chip was first demonstrated [20,76]. These early studies showed a high-speed separation of the DNA bands due to small dimensions of the channels used in the microfluidic devices. Moreover, the temperature could be rapidly changed (on the order of 100 ms), accelerating the two dozens of thermal cycles needed for the reaction [20,76]. Since the late 1990s, different immunoassays were miniaturized to be performed on microchips.

Within a couple of decades, microfluidic devices evolved to elaborated structures capable of detecting complex biological samples. From the initial implementations of PCR on-a-chip in the mid-1990s, researchers rapidly realized the potential of this new technology for diagnosis of health-related conditions, including viral infections. The increased interest to use microfluidic devices for the detection of viruses is reflected by the exponential trend in the number of papers published per year with terms as 'microfluidic', 'detection' and 'viruses' (figure 4). In most of the studied cases, new viruses have been detected with microfluidic platforms soon after the viruses first appeared (figure 4b). This prompt adaptability of miniaturized fluidic detection systems to previous outbreaks may raise hope for the accelerated development of new detectors for COVID-19 and future threads.

## (b) État des choses: real-time quantitative polymerase chain reaction and loop-mediated isothermal amplification

Detection approaches using microfluidic systems can be adapted for diagnostics of infectious diseases based on viral genetic material. The detectors based on the amplification of the genetic code require the design of specific DNA probes, called primers, that recognize a target sequence at the viral genetic material. In table 1, primers used for the detection of different viruses are presented. A successful example of the detection of the viral genetic material was presented by Fernández-Carballo et al. [5]. The team developed a PoCT qRT-PCR system for the detection of RNA-based viral pathogens [5]. As proof of concept, samples containing the Ebola virus (EBOV)

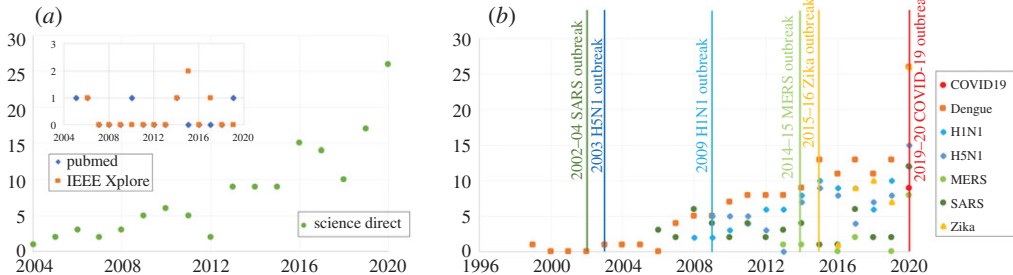

**Figure 4.** Number of publications per year in different portals for microfluidic detectors and viruses. (*a*) Searches containing all the following words: 'microfluidic devices' and 'virus detection'. (*b*) Searches in the Science Direct portal containing the words: 'microfluidic', 'detector', 'virus' and a specific disease (e.g. H1N1, Dengue, COVID19). The vertical lines indicate the time of the respective outbreaks. The data were retrieved on August 23, 2020. (Online version in colour.)

were tested with the primers described in table 1 and two master mix solutions. The group was able to achieve a processing time of 30 min, much faster than typical benchtop qRT-PCR platforms. The limit of detection of the system was about 10 RNA copies per microlitre, which is comparable to benchtop assays [5]. Similar approaches to on-chip amplification of genetic fragments were tested for the detection of avian influenza DNA (H7N9) [107], influenza A (H1N1) [85], Zika (ZIKV) [101], dengue (DENV) [96], MERS (MERS-CoV) [81], SARS (SARS-CoV) [77] and COVID-19 (SARS-CoV-2) [79] (table 1). These different studies show the great potential for microfluidics to replace or supplement benchtop assays for the diagnosis of infectious diseases.

Primers with high specificity to detect viral strains have been successfully used together with microfluidic devices to facilitate viral recognition [93,96]. For example, Yin *et al.* [96] developed a microfluidic platform for the detection of four different strains of the Dengue virus (DENV1 to 4). The high specificity of each primer allowed the identification of the four DENV and one type of Zika virus (ZIKV) [96]. In the same direction, specific primers were used to differentiate among diverse influenza strains. For the viruses tested, the detection limit varied from 40 to 3000 copies per reaction. Subtypes of influenza A, such as H1N1, H3N2, H5N1, H5N2 and H7N9; the subtypes Victoria and Yamagata of influenza B were detected using this single microfluidic device [93]. Another scenario was presented by Sabalza *et al.* [101] for the detection of different strains of the ZIKV. The research team used a region of the viral DNA conserved among the different tested strains and targeted a DNA region responsible for the formation of a virus capsid protein. Using a selected number of primers, the device detected multiple Asian-lineage strains of the ZIKV [101]. The selection of a primer for a microfluidic detector, therefore, depends on the target's structural characteristics and how wide-ranging the microdevice is designed to be.

A straightforward and cost-effective molecular technique called loop-mediated isothermal amplification (LAMP) has been increasingly used in benchtop assays and miniaturized devices for molecular diagnostics. In contrast with the different temperatures needed for each phase of a PCR cycle, LAMP is based on a single temperature—between 60°C and 65°C—for target amplification [108]. This method is compatible with reverse transcriptase and was used for SARS-CoV testing on a benchtop assay [109]. Microdevices using LAMP have been used to detect Dengue, Zika and Chikungunya [44,66,101], different subtypes of influenza A and B [51,85], MERS [81] and SARS [110]. Most recently, an RT-LAMP-based benchtop assay showed the same positive results as the WHO-approved RT-qPCR-based standard for COVID-19 diagnosis [111]. The authors suggested this new protocol could be implemented on microdevices for at-home testing. Mass at-home testing could be an effective way to prevent potential patients to leave their homes, diminishing the risk of spreading viruses such as SARS-CoV-2 [7]. The simplicity of the LAMP protocol has been considered a great advantage for its use in microfluidic devices for point-of-care testing (PoCT) [7,112]. In a final step, any of the above-mentioned variations of detecting viruses using

**Table 1.** Typical detection approaches for viruses related to recent outbreaks.

| virus | method | | ref.: |
|---|---|---|---|
| coronavirus | SARS-CoV | primer | target were two regions of SARS genome, 15240–15612 and 17743–18349. 5′-TAGGATTGCCTACGCAGACT-3′; 5′-AGAGCCATGCCTAACATGCT-3′ (for the 240 bp product), and 5′-ATTGGCTGTAACAGCTTGAC-3′ and 5′-TAG GGTAACCATTGACTTGG-3′ (for the 438 bp product), respectively. | [77] |
| | SARS-CoV-2 | antibody | SARS-CoV-2 spike antibody onto graphene | [78] |
| | SARS-CoV-2 | primer | forward 5′- CCTA CTA AAT TAA ATG ATC TCT GCT TTA CT-3′ reverse 5′- CAA GCT ATA ACG CAG CCT GTA -3′ for synthetic 22,869 nucleotides of GenBank number MN908947 | [79] |
| | MERS-CoV | primer | Orf1A protein: VIR13088F (forward) CGGCCUUCAACUGGUUGUUGUU VIR13089R (reverse) /5MAXN/AGCATAATTGTATGACCGCCAGTC N protein: VIR13090F (forward) CCUGUGUACUUCCUUCGGUACAGU VIR13091R (reverse) /5MAXN/GTAGGCATCAATATTTTGCTCAAGAAGC | [80] |
| | MERS-CoV | primer | forward: MERcv- sF_ GAGCTTAGGCTCTTTAGTAAG, reverse: MERcv- sR_ TTTTTTTTTTTGCAAATCATCTAATTAGCCTAA more sequences of primers and probes are described in the original paper | [81] |
| | MERS-CoV | primer | all the sequences are detailed in the Supplementary Material of the original article. | [82] |
| | MERS-CoV | oligonucleotide | complementary DNA 5′-CGATTATGTGAAGAG-3′; two-base-mismatch 5′-CGATTATCTGAGGAG-3′; and non-complementary DNA 5′-TTCGCACAGTGGTCA-3′ Probe: acpcPNA | [83] |
| | MERS-CoV | antibody | anti-MERS-CoV NP #20 | [84] |
| Influenza | H1N1 | aptamer | forward 5′-GGCAGGAAGACAAACA-3′ reverse 5′-ACAGCACCACAGACCA-3′ | [85–87] |
| | H1N1 | aptamer | H1N1-specific 5′-GGCAGGAAGACAAACAGCCAGCGTGACAGCGACGCGTAGGG- ACCGGCATCCGCGGGTGGTCTGTGGTGCTGT-3′ | [88] |

(*Continued.*)

**Table 1.** (*Continued.*)

| virus | method | | ref.: |
|---|---|---|---|
| H1N1 | aptamer | 5′-TTTTT TTTGG CAGGA AGACA AACAG CCAGC GTGAC AGCGA CGCGT AGGGA CCGGC ATCCG CGGGT GGTCT GTGGT GCTGT-3′ | [89] |
| H1N1 | dye tag | dye-tagged H1N1 viruses were used. | [90] |
| H1N1, H2N2, and H3N2 | dye tag | the tested viruses were tagged with fluorescence-labels. | [91] |
| H1N1 | antibody | anti-InfA nucleoprotein monoclonal antibody (A microfluidic immunomagnetic bead-based system for the rapid detection of influenza infections: from purified virus particles to clinical specimens) | [92] |
| H1N1 | antibody | a specific to influenza virus H1N1 was reported but no further detail was presented. | [65] |
| H7N9 and H9N2 | antibody | anti-H7N9 and H9N2 HA antibodies | [51] |
| H1N1, H3N2, H5N1, and H7N9; InfB; | glycan | InfA/H1N1 (105M049), InfA/H3N2 (16028588), InfA/H5N1 (Vietnam/1194/2004 NIBRG-14), InfA/H5N2 (Taiwan/duck/30-2/2005), InfA/H7N9 (Shanghai/2/2013 IDCDC-RG32A) and two strains of InfB (105M044 [Victoria strain] and 17M5012I [Yamagata strain]). sialylated N-glycans were used to group the viruses in the sample. all the sequences are detailed in the electronic supplementary material of the original article. | [93] |
| H1N1, H3N2 | RPS | geometrical characteristics and machine learning. | [94] |
| InfA and InfB | primer | the detailed sequences of primers and probes are described in the original paper. | [80] |
| HIV-1 | primer | LTR reverse 5′-CCC TGT TCG GGC GCC ACT GCT AGA GAT TTT-3′ LTR forward 5′-CCT GGG AGC TCT CTG GCT AAC TAG GGA ACC CA-3′ FAM-labelled exo probe 5′-GCT TAA GCC TCA ATA AAG CTT GCC TTG AG[T(FAM)]G-[dSpacer]T-[T(BHQ1)] CAA GTA GTG TGT GC-Spacer C3-3′ | [54] |
| | antibody | HIV1-p17 and anti-HIV1 | [42] |
| | antibody | the surface was functionalized with an array of different antigens. | [95] |

(*Continued.*)

**Table 1.** (Continued.)

| virus | | method | | ref: |
|---|---|---|---|---|
| Dengue | DENV-1 to 4 | primer and oligonucleotide | virus strains: WP-74, OBS8041, CH53489, and H241, respectively. primers and probes were not presented in the paper. | [96] |
| | | primer | detected dengue virus-specific consensus primer (DENVCP) Probe 5′- CGG TTT CTC GCG CGT TTC AGC ATA TTG A -3′ target 5′- TCA ATA TGC TGA AAC GCG CGA GAA ACC G -3′ | [58] |
| | | antibody | the use of highly specific monoclonal antibodies (MAB$_D$) against each of the four dengue virus serotypes was reported but not further detail was presented. | [23] |
| | DENV | antibody | P6-NS1 and P4-IgG | [97] |
| | | antibody | the surface was functionalized with anti-NS1 for detection of NS1 DENV | [98–100] |
| | | antibody | different variants of NS1 | [47] |
| | | antibody | silica beads were functionalized with anti-flavivirus 4G2 | [56] |
| | | SS RNA | single-strand DNA probes were used, no further details were presented. | [44] |
| Zika | ZIKV | primer | virus strains: NR-50240, NR-50242, NR-50244, NR-50358, PRVABC59, R103451, and PLCal_ZV. the detailed sequences of primers and probes are described in the original paper. | [101] |
| | | primer | lineage of the zika virus (East African) NC_012532.1. forward 5′ AGACTTATGGTTGTTGAGGAAGCC 3′ (24 bp) reverse 5′ CGCCATTCGTTTGAGTCTATCCC 3′ (23 bp) probe 5′ -HS-(CH2)6- AGACTTATGGTTGTTGAGGAAGCC 3′ (24 bp) all the sequences are detailed in the Supplementary Material of the original article. | [102] |
| | | primer | forward outer 5′-CGGATGGGATAGGCTCAAAC-3′ backward outer 5′-ATGGACCTCCCGTCCTTG-3′ forward inner 5′- CCTGAGGGCATGTGCAAACCTAGAATGGCAGTCAGTGGAGAT-3′ backward inner 5′-ACCCTCAACTGGATGGGACAACTGGAGCTTGTTGAAGTGGTG-3′ forward loop 5′-CATCAATTGGCTTCACAACGC-3′, backward loop 5′- GGGAAGAAGTTCCGTTTTGCTC-3′ | [45] |

**11**

(*Continued.*)

**Table 1.** (*Continued.*)

| virus | method | | ref.: |
|---|---|---|---|
| | four-way junction | the surface was functionalized with Universal DNA-Hairpin (UDH) probe. Both synthetic ZIKV ssDNA strand (138 nt) and ZIKV RNA amplicon (~147 nt) were tested. | [103] |
| | SS RNA | single-strand DNA probes were used, no further details were presented. | [44] |
| | antibody | the surface was functionalized with protein ZIKV E for detection of anti-ZIKV E antibody. | [104] |
| | antibody | different variants of NS1 | [47] |
| Ebola | Primer | forward primer [A/5'-CTA CTG TAT TTC ATA AGA AGA GAG TTG AAC C-3'/5'-AAT TGT TGT TCT ACT GAT CCA CAA GTC TTA C-3'/5'-ATA TGT CCG ACC TTG AAA AAA GGA TTT TTG [FAM-dT][THF][BHQ1-dT] GAC AGT AGT TTT TGC [3'-phosphate]/160 bp] reverse primer [B/5'-CTA CTG AGT CCA GTA TAG AGT CAG AAA TAG TA-3'/5'-CTG AGT TGT TAA GAA TAA TCT CAA TTT GGT-3'/5'-AAT GAC TAC TCC TAG GAT GCT TCT ACC TGT [FAM-dT][THF][BHQ1-dT] GTC AAA ATT CCA TAA [3'-phosphate]/127 bp] probe [C/5'-GAC GAC AAT CCT GGC CAT CAA GAT GAT GAT CC -3'/5'-CGT CCT CGT CTA GAT CGA ATA GGA CCA AGT C -3'/5'-GAT GAT GGA AGC TAC GGC GAA TAC CAG AG [FAM- dT] T [THF] C [BHQ1-dT] CGG AAA ACG GCA TG [3'-phosphate]/168 bp] | [51] |
| EBOV | primer | forward 5'-GTCCGTCGTTCCAGTCATTT-3' reverse 5'-CCCTCTTGGATGCTGAGTTA TG-3' fluorogenic probe 5'-TAAGTGACTCTGCT TGCGGTACAGC-3' | [5] |
| | primer | used CRISPR-Cas13a CRISPR RNA 5'- GGGGAUUUAGACUACCCAAAAACGAAGG-GGACUAAAACGUGGCGUCAUCUCCAGCCUUAUCAUG target RNA 5'GAACAUUGAUAAGGCUGGAGAUGACGCC-ACAACAGCUUUGUGAGCUAUUUUCCAUUCAAAAACACU-GGGGGCAUCCUGUGCUACAUAGUGAAACAGCAAU -3' | [64] |
| | antibody | the surface was functionalized with anti-EBOV | [105,106] |
| Chikungunya | oligonucleotide | capture DNA probe 5'-NH2- TGC TCC GCG TCC TTT ACC AA -3' target DNA 5'- TTG GTA AAG GAC GCG GAGCA -3' non-complementary DNA 5' - CTA TGC TTA CAC GTA GAC TGT GC -3' | [46] |
| CHIKV | SS RNA | single-strand DNA probes were used, no further details were presented. | [44] |

amplified biological assays require an interface to the end-user, i.e. a visualization method to indicate a detection event. One typical way is to use gel electrophoresis, which will deliver stained patterns such as the ones known from benchtop devices. Further details are given below, in the chapter on User Interfaces.

## (c) Looking for proteins instead of genetic material

Unlike the above-mentioned approaches to detect viruses based on their genetic material in PCR or LAMP protocols using *primers*, the detection of viruses based on their viral proteins is usually performed with the use of *antibodies*. Antibodies are protective proteins produced by the immune system in response to the presence of a foreign substance, called an antigen (as defined by the Encyclopedia Britannica). Antibodies are generally Y-shaped and have very specific structural characteristics that enable binding only with unique targets. This property is the reason antibodies are widely applied in detectors for biomolecules [98,105,113–117]. An example of a microfluidic detector using antibodies against viral proteins was described by Wang *et al*. [92]. They developed a device capable of recognizing two subtypes of Influenza A (Inf.-A) simultaneously, based on antibodies immobilized on magnetic beads [92]. Antibodies anti-H7 and anti-H9 were used to detect H7N9 and H9N2 Inf.-A subtype viral proteins in concentrations as low as 3.4 and 4.5 ng ml$^{-1}$, respectively [92]. Microfluidic devices can also use aptamers[3] to screen for viruses. Detection of Inf.-A, for example, was possible by using a combination of antibodies and aptamers. The subtypes of the inf.-A virus are characterized by a combination of haemagglutinin (HA) and neuraminidase (NA) proteins [86,92,119]. HA is a viral surface protein that participates in the viral invasion and propagation process in host cells [92,119]. Anti-HA antibodies were used to capture the viruses, and the suitability of several aptamers against H1N1 was tested. The aptamers sensitive to Inf.-A were then tested against Inf.-B, and only those uniquely specific to H1N1 were selected to be used in the device [86]. Specific aptamers can be used to capture the virus, which is then exposed to a primary antibody that binds to the virus surface. The primary antibody enables the linkage to a secondary antibody with a fluorescent element, therefore generating an optically detectable signal [89].

When fighting against pathogens such as viruses, the human body produces several antibodies that will circulate the blood system and potentially neutralize the infectious agent. Another way to detect infections, instead of directly looking for the pathogenic agent, e.g. using antibodies introduced by the detector as described above, is to detect the presence of those antibodies already present in the system of patients [120]: in a method called serological testing, detectors for different diseases use the presence of specific antibodies in the patient's serum as biomarkers, [42,84,104,113,120,121]. Microfluidic devices using serological testing have been developed to detect ZIKV, MERS-CoV and DENV. This is therefore also a potentially promising approach to detect SARS-CoV-2 [74]. Recently, Wang et al. [104] presented a capacitive sensor that was evaluated using immobilized ZIKV. Serum samples were tested for the detection of anti-Zika antibodies and the detection happened in concentrations as small as 10 molecules in 30 µl [104]. Similarly, Hoy *et al*. [84] created a microfiber-based detector for MERS-CoV. The His-MERS-NP antigen protein was immobilized at polystyrene microfibres and exposed to a commercial anti-MERS-NP. Although tests with patient samples are still to be done, the detection limit using a buffer solution was quantified to be 200 µg ml$^{-1}$ [84]. Another detection approach was followed by Cecchetto *et al*. [98] who developed a Dengue capacitive detector based on non-structural protein 1 (NS1) [98]. The NS1 is a viral glycoprotein secreted by cells [47]. This protein has been considered a biomarker for early-stage Dengue with concentrations of 0.04– 2.00 µg ml$^{-1}$ in serum-infected patients [100]. In their most recent publication, the group presented a device with a ferrocene-tagged, peptide-modified surface that is equipped with anti-NS1. The presence of the NS1 in the samples was detected by changes in the capacitance of the electrode [98]. In

[3]Aptamers are short single-stranded sequences of nucleotides (DNA or RNA) that can bind to specific targets ranging from simple inorganic molecules to large protein complexes, or even whole cells. Although they may occur naturally, most are synthesized and selected *in vitro* for definite targets [118].

a final step, any of the above-mentioned variations of detecting viruses by linking antibodies to the target particles requires an interface to the end-user, i.e. a visualization method to indicate a detection event. One typical way is to use fluorescent markers that get bound to the target particle and will generate an optical response. Other approaches use electrochemical effects to, e.g. detune electrical properties of electrodes or transistors, in the wake of binding molecules to a functionalized surface or volume. Further details are given below, in the section on User Interfaces.

### (d) A detector without biological ligands

Instead of applying the same detection approaches used in benchtop assays, some microfluidic detectors explore the resistive pulse sensing (RPS) principle [70,94,113,122–124]. RPS detection is based on effects associated with the translocation of molecules through pores, rather than on chemical bonding as in the previously discussed detection principles. The first description of this technique is attributed to Wallace Coulter in 1953. Since then, the method has been substantially improved and is widely used for counting cells [125,126]. A resistive pulse sensor, often referred to as a Coulter counter, consists of two reservoirs with their respective electrodes and is filled with a conductive solution (electrolyte). The key component is a small pore (or channel) that connects both reservoirs. The electrodes set up an electric field, which results in an ionic current across the pore. This electric current is dependent on the flow of substances through the pore upon translocation of biomolecules of interest (target elements) through the pore, a temporary rapid increase in the electric resistance occurs. This is the signal that can get evaluated for detection, and which coined the term resistive pulse sensor. In recent decades, micro- and nanofabrication technologies have allowed minimizing the diameter of the pore, creating devices with potentially single-molecule sensitivity [70,124,127,128]. RPS-based detectors were used for single-virus identification for the diagnosis of influenza, with a discriminability above 95% between H1N1 and H3N2 when analysing the detection pattern for 20 virions or more [94]. RPS was also previously used to study the HIV-1 and hepatitis B virus [62,129]. This technique can be modified in a variety of ways, for example with the use of multi-pore membranes to allow for concurrent detections [130]. This technique is highly sensitive, and the design of the device is simple. Fabrication, however, can be challenging, and the purification of samples imposes a major obstacle for the translation of such devices into PoCT [131].

## 5. User interface: signs of detection in microfluidic systems

In the previous section, some approaches for the detection of viruses based on biochemical (e.g. DNA/RNA with primers and antigens with antibodies) and biophysical (e.g. RPS) processes were introduced. In this section, the methods used to indicate the detection to the user, such as fluorescence proteins, optical inspection of PCR bands and peaks in the electric current are discussed.

### (a) Fluorescent markers

It is a common practice to use fluorescent elements to indicate the detection of a target. These elements can be added to the device [63,89,93], attached to the target element [84,90,132] or replicated during amplification steps [5,101,133]. Examples of fluorescent elements used in microfluidics are cyanine-5 (Cy5), methylene blue (MB) [79], fluorescein [77], fluorescein amidites (FAM), hexachlorinated fluorescein (HEX) [110], SYTO-82 [134], AF 488 [56], EvaGreen and SYBR Green [81,107]. Fluorescence is a very versatile method that can be used in different materials (e.g. paper [41], polymers [63] and structure-free [89]). It can also be adapted to automated analysis via smartphones or desktop scanners [107,135]. The adaptability of fluorescence methods to smartphones has been helping in the detection of viruses such as H7N9 avian influenza and H1N1 [87,107]. Shi *et al.* [107] developed a microfluidic device for the detection of the genomic DNA of

H7N9 avian influenza. Their system consists of a device for amplification of the target DNA, a bluish light-emitting diode and a narrowband filter. A smartphone can be attached to record the fluorescent signals generated by different dyes (EvaGreen, SYBR Green and FAM fluorescent probe) in real-time [107]. Another system that allows viral detection with the help of a smartphone was presented by Ma *et al.* [87]. The group developed a method capable of processing a sample-to-answer for the detection of the H1N1 virus in less than 40 min. Their system was based on colorimetric LAMP, a passive microfluidic device and an optical detector. A smartphone was used to control the system and analyse the results. Handheld detector systems have the potential to be used as PoCT afar form laboratories or healthcare facilities [45,55,136].

## (b)  Colorimetric methods

Variations of colorimetric methods which apply modifications of surfaces with chemical components can also be used to indicate the detection of viruses to the user. In these systems, colorimetric signals are caused by coloured immunoreaction labels such as colloidal nanoparticles of gold, carbon and selenium [16]. Teengam *et al.* [83] used the aggregation of gold and silver nanoparticles to change the colour of their paper wells upon detection of the MERS-CoV and other pathogens. The observed detection limit for MERS-CoV was 1.53 and 1.3 nM for other tested pathogens [83]. Without using fluorescence elements, the group was able to image the results with a regular scanner and to perform the analysis using the software ImageJ (National Institutes of Health, USA). The visible difference upon detection was based on the action of peptide nucleic acids, which were responsible for the aggregation of the nanoparticles and the target DNA binding. Another paper-based detector was presented by Aydin *et al.* [137] for the detection of Hepatitis B viral DNA. The team used conjugated polyelectrolytes which changed colour upon binding of primer and target DNA. In this device, polyvinylidene fluoride (PVDF) membranes were deposited with poly[N,N,N-triethyl-3-((4-methylthiophen-3-yl)oxy)propan-1-aminium bromide] which would be read as orange or pink/grey depending on the conformation between the target and the surface. The detection limit was reported to be 1 nM for clinical samples [137]. This type of detector has the advantage of emitting signals in the visible range, leading to rapid identification of the result without the need for extra processing steps involving UV-light.

## (c)  Electrochemistry-based detectors

Electrochemical methods can use variations of the impedance between electrodes to indicate a detection without using extra reagents as fluorescent labels. Different electrochemical assays can be used to monitor the surface or the volume between electrodes while testing a sample. A variety of detectors have been proposed based on the functionalization of the surfaces of electrodes for specific binding of target molecules [46,57,61,98,99,102,103,138,139]. As an example, surface functionalization was used for the detection of DENV based on the immobilization of NS1 antigens on the surface of gold electrodes [98]. The bonding between antigen and antibody causes a shift in the impedance (figure 5*a*) and indicates the presence of the virus in the patient's body. The detection is visualized in Nyquist or Bode diagrams and requires spectroscopy equipment [57,65,99,139]. Another example of functionalized electrodes was used for detecting ZIKV through the immobilization of DNA-hairpin probes on gold discs to capture Zika RNA in the samples [103]. The immobilization process usually requires the use of thiol self-assembled monolayers (SAMs) or similar functional groups [57,142]. The functionalized electrodes that allow for the binding of the viral RNA to detect ZIKV were coupled with a four-way junction sensor. A four-way junction sensor is a multi-component sensing element designed to specifically hybridize with the target genetic material providing a measurable signal [143,144].

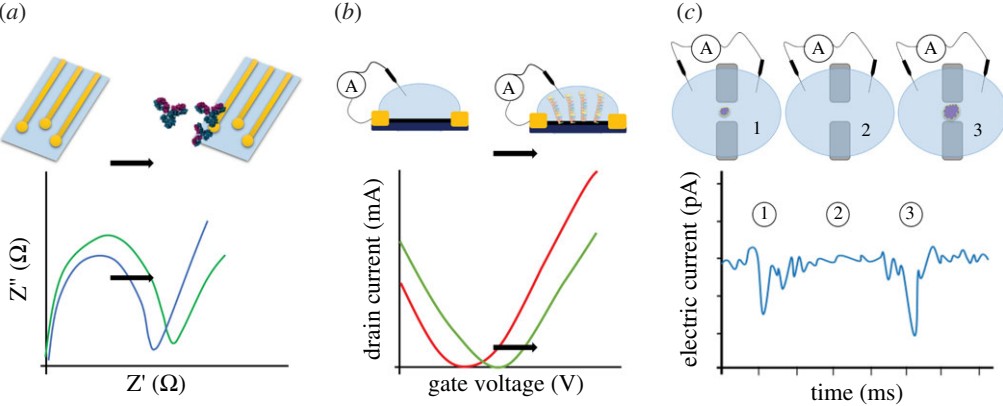

**Figure 5.** Visualization of detection events in biomolecule detectors based on electric signals. From left to right: (*a*) impedance spectroscopy (applied in electrochemistry-based detectors where, e.g. antibodies bind to target material), (*b*) transfer function (applied in immunoFET devices where target particles in a solution induce electrostatic perturbations) and resistance pulses (applied in resistive pulse sensors where target particles lead to temporary current blockages). Images were adapted from [99,140,141]. (Online version in colour.)

## (d) Electric current-based signalling

Field-Effect Transistors (FET) and RPS are label-free detectors that use the electric current to indicate detection. FET-based detectors—also referred to as immunoFET devices—are used for immune assays and early diagnosis. In these devices, the interactions between the binding elements immobilized on the surface and the target particles in a solution induce electrostatic perturbations that alter the voltage threshold of the transistor [114]. The curves of the characteristic transfer function show a shift upon antigen-antibody binding in comparison to non-specific antigens [105,106,116] (figure 5*b*). FET-based sensors were described to detect the Ebola virus through the immobilization of Anti-EBOV antibodies on the surface of the device and by monitoring the electric current [105]. Recently, Seo *et al*. [145] presented a graphene-based FET for the detection of SARS-CoV-2 using immobilized spike antibodies. The group was able to achieve a detection limit of $2.42 \times 10^2$ copies per ml of clinical samples. RPS is another class of label-free detectors in which the variation of the electric current through a pore can be used to indicate the detection of viruses [74]. Tsutsui *et al*. [130] used an RPS with a nanopore with a diameter of 300 nm to detect and discriminate between different influenza viruses [130]. For the detection of nanoparticles with 200 nm diameter, the team reported spikes in the electric current on the order of 1 nA. An example of an electric current time signal for detection events using RPS is represented in figure 5*c*. Label-free detectors can reduce the number of required reagents. Some variations even require no biochemical elements at all. In emergency fabrication situations, the ability to perform specific detections without the need for extra biotechnological products for labelling may be considered advantageous [74].

## 6. Conclusion and perspectives

Within about half a year, COVID-19 has spread globally, causing hundreds of thousands of fatalities. The current scenario has put immense pressure on healthcare and economic systems worldwide. The sanitary situation is unparalleled in recent years. It has called for far-reaching restrictions in people's lives, such as social distancing requirements, work, school and travel restrictions, and local shutdowns. It has also led to a spike in demand for hygiene items such as 70% alcohol, personal protective equipment (PPE) such as masks and face shields, mechanical respirators and screen test kits. At the same time, we have witnessed an extraordinary effort

from small and large companies, universities and common citizens to find ways to help. Despite of, or even because of, all these efforts, a critical bottleneck in this pandemic became evident: around the globe, there is a lack of reliable screen test kits for a new virus to prevent the unrestrained spreading of a disease such as COVID-19. Therefore, the ability to rapidly produce test kits for quick diagnosis seems to have become instrumental to avoid further social and economic disruptions while facing new varieties of contagious diseases. In the present work, we have discussed some of the most recent advancements in microfluidic-based detectors for viruses. The devices reviewed are based on a variety of approaches and methods but are all capable of detecting and discriminating viral targets in real test samples from medical patients. Many highly efficient microfluidic devices have taken advantage of well-established benchtop assays (i.e. implemented in macroscopic analysis tools rather than miniaturized, portable devices) and adapted them to miniaturized lab-on-a-chip versions. This is possible because recent concentrated research efforts on the SARS-CoV-2 virus have unveiled its genetic material, the proteins and other molecules that form the virus, and the memory antibodies for the disease [146–148]. These discoveries allowed for the development of new microfluidic detectors based on biological ligands. One such category of detectors uses *primers* as biological ligands when applying protocols involving the amplification of *genetic material* (e.g. DNA, RNA), such as polymerase chain reaction (PCR) and loop-mediated isothermal amplification (LAMP). Another such category uses *antibodies* as biological ligands to bind target particles for the detection of viruses based on their *viral proteins* [79,145,149]. Besides, other innovative solutions to detecting viruses exploit fabrication capabilities of micro- and nanotechnology to detect viruses with fewer or even no biological ligands but rather based on their geometry or electrical properties. Resistive pulse detectors are an example of this category of devices. The different detection approaches are compatible with different visualization methods to indicate detection events. Some common methods are based on fluorescent markers, on colorimetric approaches, on electrochemically induced shifts in the spectroscopic response, on modified transfer functions of involved transistors, or on modified current signals. The combination of the various detection and visualization/interface methods allows for a range of sensitivities and resolution limits, but also strongly varies the requirements for device complexity and supporting biochemical supplies. A common guideline for the commercial success of micro- and nanotechnology products claims that 'simpler is better'. This attitude likely bears additional value when rapid development of new detector devices and fast fabrication are needed, such as under emergency fabrication requirements as in the situation imposed by a pandemic such as COVID-19. In this context, paper-based devices offer unique characteristics of the ubiquitous availability of material and relatively easy fabrication and disposal processes. This might be the simplest kind of a microfluidic device that can be rapidly adapted, and mass-produced for home testing populations. However, even those paper-based devices still require primers or antibodies for detection, which might constitute a problematic obstacle for developing, emerging or low gross domestic product countries [150]. The same dependency on biotechnological material is required for electrochemistry-based detectors, which may additionally rely on large benchtop equipment such as spectrometers. An electrochemistry-based detection method that does not require much additional equipment or supplies is based on FET (immunoFET devices). The fabrication of sophisticated FET-based detectors, however, can be quite challenging and may not be feasible in many regions of the globe. New protocols to create micro and nanopores for RPS-based devices fall into the same category as the FET devices mentioned above – elimination of the need for biological ligands at the expense of much-increased fabrication challenges. The sensitivity of RPS detectors to the size and shape of the target particle eliminates the need for primers or antibodies, but there is still the need for highly complex fabrication capabilities and very sensitive amperemeters. In summary, a large variety of methods and approaches can be applied for the detection of viruses using microfluidic devices. They much vary in the complexity of the fabrication process, in the need for biochemical supplies and ancillary instruments, and detection properties. Therefore, no single approach stands out as a best-fit solution. Instead, a selection from the presented alternatives, or

additional new ones potentially getting developed, should be made based on the complexity of the detection and discrimination of the specific virus of interest and based on local community needs and, cost considerations, and delivery timelines to find an adequate solution for producing screening tests and making them widely available.

Data accessibility. This article does not contain any additional data.

Authors' contributions. J.A.B. conceived of the study, designed the study, carried out the literature review, carried out the design of the figures and drafted the manuscript; R.G.M. participated in the design of the study and drafted the manuscript; S.A. coordinated the study and critically revised the manuscript. All authors gave final approval for publication and agree to be held accountable for the work performed therein.

Competing interests. We declare we have no competing interests.

Funding. We acknowledge the support of the Natural Sciences and Engineering Research Council of Canada (NSERC).

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
