## [Reviewer comments · Proceedings. Mathematical, Physical, and Engineering Sciences]

Review History

RSPA-2020-0398.R0 (Original submission)

Review form: Referee 1

Is the manuscript an original and important contribution to its field?

Excellent

Is the paper of sufficient general interest?

Excellent

Is the overall quality of the paper suitable?

Excellent

Can the paper be shortened without overall detriment to the main message?

Yes

Do you think some of the material would be more appropriate as an electronic appendix?

No

Do you have any ethical concerns with this paper?

No

Recommendation?

Accept with minor revision (please list in comments)

Comments to the Author(s)

This paper reviewed the recent developments on microfluidic devices for viruses detection. Authors discussed different detection approaches, methods of signalling and fabrication techniques. This review paper is very informative and helpful for readers to chose and develop suitable miniaturized detectors for specific virus detection tasks. I would like to highly recommend this review article to be accepted by the Journals of Proceedings A after a optional minor revision.

- 1) Page 2, lines 43-49, Introduction, I would suggest authors to update the the infection cases and deaths of US and worldwide according to the latest information;
- 2) Page 3, line 34, "Many microfluidic devices are been...", here, I would suggest to replace the word "are" by "have";
- 3) Page 19, line 34, "References sources not found", authors need to check the correct the citation here.

Review form: Referee 2

Is the manuscript an original and important contribution to its field?

Good

Is the paper of sufficient general interest?

Good

Is the overall quality of the paper suitable?

Acceptable

Can the paper be shortened without overall detriment to the main message?

Yes

Do you think some of the material would be more appropriate as an electronic appendix?

No

Do you have any ethical concerns with this paper?

No

Recommendation?

Major revision is needed (please make suggestions in comments)

Comments to the Author(s)

Dear authors,

I have read your article. Of course the topic of the review is of interest to not only research community but also for general public. To make it widely acceptable and easy for readers to understand, the article should undergo several structural changes (see my comment below):

1. The review is on: microfluidic devices for viruses detection. However, the authors have spent the whole of 1st paragraph (introduction section) only on coronavirus. Although it is interesting but there is no need to focus that much on this topic alone. Therefore I strongly suggest the authors to cut short this to not more than 4-5 lines.
2. Again in the section of "Recent outbreaks", a lot of information is provided in paragraph 2 (page 6). All this information can be found in various relevant articles. I suggest the authors to cut short this to not more than 4-5 lines.

3. Figure 1: the authors say that “searches containing the word: microfluidic, devices, virus and detection. Of course the numbers will be high (it is expected as it covers different topics). I think as this review is on “microfluidics for virus detection”. This figure should only represent the list of publications dealing with “Microfluidics for virus detection” as such. Then we will understand the trend how microfluidics are used for virus detection. Otherwise this is a misleading figure. The figure should be remodified.

4. P21, L34: analysis (see Figure 3Error! Reference source not found). Please check this.

5. I find the organization of the article should be better. The review should be easy to follow in its structure. The authors need to restructure the article. For e.g. they should have discussed about the microfluidic fabrication etc. after the section of “Recent outbreaks”. Then they should talk about detection of various viruses using microfluidic devices. In this they should have categorized papers more clearly with various subheading. Then finally come to conclusion.

6. In terms of conclusion: I feel they should have discussed more scientific conclusions for various virus detection using microfluidics then how they perceive some of the potential possibilities for COVID19 detection from the existing literature etc. The purpose of this review is to show way for researchers to exploit the microfluidic devices for virus detection and especially in current scenario the COVID19 detection. This is missing in the current conclusion section

7. General comment: Grammar should be crosschecked

Decision letter (RSPA-2020-0398.R0)

07-Aug-2020

Dear Dr Berkenbrock

The Reviews Editor of Proceedings A has now received comments from referees on the above paper and would like you to revise it in accordance with their suggestions which can be found below (not including confidential reports to the Reviews Editor).

Please submit a copy of your revised paper within four weeks - if we do not hear from you within this time then it will be assumed that the paper has been withdrawn. In exceptional circumstances, extensions may be possible if agreed with the Editorial Office in advance.

Please note that it is the editorial policy of Proceedings A to offer authors one round of revision in which to address changes requested by referees. If the revisions are not considered satisfactory by the Reviews Editor, then the paper will be rejected, and not considered further for publication by the journal. In the event that the author chooses not to address a referee's comments, and no scientific justification is included in their cover letter for this omission, it is at the discretion of the Editor whether to continue considering the manuscript.

- Acknowledgements
- Funding statement

To revise your manuscript, log into <http://mc.manuscriptcentral.com/prsa> and enter your Author Centre, where you will find your manuscript title listed under "Manuscripts with Decisions." Under "Actions," click on "Create a Revision." Your manuscript number has been appended to denote a revision.

You will be unable to make your revisions on the originally submitted version of the manuscript. Instead, revise your manuscript and upload a new version through your Author Centre.

When submitting your revised manuscript, you will be able to respond to the comments made by the referee(s) and upload a file "Response to Referees" in "Section 6 - File Upload". Please use this to document how you have responded to the comments, and the adjustments you have made. In order to expedite the processing of the revised manuscript, please be as specific as possible in your response to the referee(s).

IMPORTANT: Your original files are available to you when you upload your revised manuscript. Please delete any unnecessary previous files before uploading your revised version.

When revising your paper please ensure that it remains under 28 pages long. Your paper has been ESTIMATED to be 22 pages.

Once again, thank you for submitting your manuscript to Proc. R. Soc. A and I look forward to receiving your revision. If you have any questions at all, please do not hesitate to get in touch.

Yours sincerely
Raminder Shergill
proceedingsa@royalsociety.org

on behalf of
Professor Michel Destrade
Reviews Editor
Proceedings A

Reviewer(s)' Comments to Author:

Referee: 1

Comments to the Author(s)

This paper reviews recent developments on microfluidic devices for viruses detection. The authors discuss different detection approaches, methods of signalling and fabrication techniques. This review paper is very informative and helpful for readers to chose and develop suitable miniaturized detectors for specific virus detection tasks. I would like to highly recommend this review article to be accepted by Proceedings A after minor revisions.

- 1) Page 2, lines 43-49, Introduction, I suggest that the authors update the infection cases and deaths of US and worldwide according to the latest information;
- 2) Page 3, line 34, "Many microfluidic devices are been...", here, I would suggest to replace the word "are" by "have";
- 3) Page 19, line 34, "References sources not found", authors need to check the correct the citation here.

Referee: 2

Dear authors,

I have read your article. Of course the topic of the review is of interest to not only to the research community but also to the general public. To make it widely acceptable and easy for readers to understand, the article should undergo several structural changes, see my comments below.

1. The review is on 'Microfluidic devices for viruses detection'. However, the authors spend the whole 1st paragraph (Introduction section) on the coronavirus only. This is interesting but there

is no need to focus that much on this topic alone. Therefore I strongly suggest that the authors shorten this part to not more than 4-5 lines.

2. In the section on “Recent outbreaks”, a lot of information is provided in paragraph 2 (page 6). All this information can be found in various relevant articles. I suggest that the authors shorten this part to not more than 4-5 lines.

3. Figure 1: the authors mention “searches containing the word: microfluidic, devices, virus and detection”. Of course the search numbers will be high: this is expected because the search covers different topics. This review is on “microfluidics for virus detection”, and the search should be on this theme, within quotes. The figure should only represent the list of publications dealing with “Microfluidics for virus detection” as such. Then we will understand the trends on how microfluidics are used for virus detection. Otherwise a misleading figure is presented and it should be remodified.

4. P21, L34: analysis (see Figure 3Error! Reference source not found). Please check this.

5. I find the organization of the article could be improved. The review should be easy to follow in its structure and the authors need to restructure the article. For example, they should discuss the microfluidic fabrication etc. after the section on “Recent outbreaks”. Then, they should talk about the detection of various viruses using microfluidic devices. In that part, they should sort papers more clearly with into categories with clear subheading. Then finally they can come to conclusion.

6. About the conclusion: I feel the authors should have presented and discussed more scientifically-minded conclusions for various virus detection modes using microfluidics; and then explain how they perceive some of the potential possibilities for COVID19 detection from the existing literature, etc. Indeed, the purpose of this review is to show ways for researchers to exploit microfluidic devices for virus detection and especially in the current scenario of COVID19 detection. This aspect is missing in the current conclusion section

7. General comment: The grammar should be checked throughout.

Author's Response to Decision Letter for (RSPA-2020-0398.R0)

See Appendix A.

RSPA-2020-0398.R1 (Revision)

Review form: Referee 1

Is the manuscript an original and important contribution to its field?

Good

Is the paper of sufficient general interest?

Excellent

Is the overall quality of the paper suitable?

Good

Can the paper be shortened without overall detriment to the main message?

Yes

Do you think some of the material would be more appropriate as an electronic appendix?

No

Do you have any ethical concerns with this paper?

No

Recommendation?

Accept as is

Comments to the Author(s)

Authors have addressed all my comments properly.

Review form: Referee 2

Is the manuscript an original and important contribution to its field?

Excellent

Is the paper of sufficient general interest?

Excellent

Is the overall quality of the paper suitable?

Good

Can the paper be shortened without overall detriment to the main message?

Yes

Do you think some of the material would be more appropriate as an electronic appendix?

No

Do you have any ethical concerns with this paper?

No

Recommendation?

Accept as is

Comments to the Author(s)

The authors have done the necessary changes to the manuscript.

Decision letter (RSPA-2020-0398.R1)

06-Oct-2020

Dear Dr Berkenbrock

On behalf of the Reviews Editor, I am pleased to inform you that your manuscript entitled "Microfluidic devices for the detection of viruses: aspects of emergency fabrication during the

COVID19 pandemic and other outbreaks" has been accepted in its final form for publication in Proceedings A.

COVID-19 rapid publication process: We are taking steps to expedite the publication of research relevant to the pandemic. If you wish, you can opt to have your paper published as soon as it is ready, rather than waiting for it to be published on the scheduled Wednesday.

This means your paper will not be included in the weekly media round-up which the Society sends to journalists ahead of publication. However, it will appear in the COVID-19 Publishing Collection which journalists will be directed to each week

(<https://royalsocietypublishing.org/topic/special-collections/novel-coronavirus-outbreak>)

If you wish to have your paper published immediately please notify production@royalsociety.org and press@royalsociety.org.

Our Production Office will be in contact with you in due course. You can expect to receive a proof of your article soon. Please contact the office to let us know if you are likely to be away from e-mail in the near future. If you do not notify us and comments are not received within 5 days of sending the proof, we may publish the paper as it stands.

Under the terms of our licence to publish you may post the author generated postprint (ie. your accepted version not the final typeset version) of your manuscript at any time and this can be made freely available. Postprints can be deposited on a personal or institutional website, or a recognised server/repository. Please note however, that the reporting of postprints is subject to a media embargo, and that the status the manuscript should be made clear. Upon publication of the definitive version on the publisher's site, full details and a link should be added.

You can cite the article in advance of publication using its DOI. The DOI will take the form: 10.1098/rspa.XXXX.YYYY, where XXXX and YYYY are the last 8 digits of your manuscript number (eg. if your manuscript number is RSPA-2017-1234 the DOI would be 10.1098/rspa.2017.1234).

For tips on promoting your accepted paper see our blog post:
<https://royalsociety.org/blog/2020/07/promoting-your-latest-paper-and-tracking-your-results/>

Thank you for your submission. On behalf of the Editors of the journal, we look forward to your continued contributions to the Journal.

Best wishes
Raminder Shergill,
Proceedings A Editorial Office
proceedingsa@royalsociety.org

on behalf of
Professor Michel Destrade
Reviews Editor
Proceedings A

Reviewer(s)' Comments to Author:

Referee: 1

Comments to the Author(s)
Authors have addressed all my comments properly.

Referee: 2

Comments to the Author(s)

The authors have done the necessary changes to the manuscript.

Appendix A

Reviewer(s)' Comments to Author:

Referee: 1

Comments to the Author(s)

This paper reviews recent developments on microfluidic devices for viruses detection. The authors discuss different detection approaches, methods of signalling and fabrication techniques. This review paper is very informative and helpful for readers to chose and develop suitable miniaturized detectors for specific virus detection tasks. I would like to highly recommend this review article to be accepted by Proceedings A after minor revisions.

1) Page 2, lines 43-49, Introduction, I suggest that the authors update the infection cases and deaths of US and worldwide according to the latest information;

Authors: Thank you for your comment. We included the most recent data as suggested. We used the numbers available for the day before the submission of this reviewed version.

2) Page 3, line 34, "Many microfluidic devices are been...", here, I would suggest to replace the word "are" by "have";

Authors: Thank you, we replaced it as suggested.

3) Page 19, line 34, "References sources not found", authors need to check the correct the citation here.

Authors: Thank you for noticing that. We apologize for this error on cross-referencing the figure 3. We corrected the linkage.

Referee: 2

Dear authors,

I have read your article. Of course the topic of the review is of interest to not only to the research community but also to the general public. To make it widely acceptable and easy for readers to understand, the article should undergo several structural changes, see my comments below.

1. The review is on 'Microfluidic devices for viruses detection'. However, the authors spend the whole 1st paragraph (Introduction section) on the coronavirus only. This is interesting but there is no need to focus that much on this topic alone. Therefore I strongly suggest that the authors shorten this part to not more than 4-5 lines.

Authors: Thank you for you comment. We carefully considered the suggestion and shortened the mentioned paragraph. The authors also agreed and included the term COVID19 in the title since the focus of this review had been positively indicated by the editor prior to the writing of the manuscript.

2. In the section on “Recent outbreaks”, a lot of information is provided in paragraph 2 (page 6). All this information can be found in various relevant articles. I suggest that the authors shorten this part to not more than 4-5 lines.

Authors: Thank you for your suggestion. We re-worked the paragraph.

3. Figure 1: the authors mention “searches containing the word: microfluidic, devices, virus and detection”. Of course the search numbers will be high: this is expected because the search covers different topics. This review is on “microfluidics for virus detection”, and the search should be on this theme, within quotes. The figure should only represent the list of publications dealing with “Microfluidics for virus detection” as such. Then we will understand the trends on how microfluidics are used for virus detection. Otherwise a misleading figure is presented and it should be remodified.

Authors: Thank you for raising this concern. We reviewed the values and updated the graph. We re-phrased the description of the figure to clarify how the search was performed. In brief, the search terms were: “microfluidic device” AND “virus detection”. The search mechanism then looks for works/articles that contain the terms ‘microfluidics devices’ + ‘virus detection’, as a group of words connected by an AND clause. The graphs DO NOT represent a search where the terms were assembled by an OR clause. The use of quotes (e.g., “microfluidics for virus detection”) is overly restrictive and resulted a single review paper (search performed at Science Direct portal in August 2020).

4. P21, L34: analysis (see Figure 3Error! Reference source not found). Please check this.

Authors: Thank you for noticing that. We apologize for this error on cross-referencing the figure 3. We corrected the linkage.

5. I find the organization of the article could be improved. The review should be easy to follow in its structure and the authors need to restructure the article. For example, they should discuss the microfluidic fabrication etc. after the section on “Recent outbreaks”. Then, they should talk about the detection of various viruses using microfluidic devices. In that part, they should sort papers more clearly with into categories with clear subheading. Then finally they can come to conclusion.

Authors: Thank you for your comment. The changes in the structure were made as suggested. We agreed that the fabrication of the device logically precedes the detection and the signal visualization. On the other hand, the authors had considered the wider readership of the Proceedings A would be interested first on detection approaches and only those inclined to production processes would be interested on the fabrication details. Keeping this mindset, the theme was introduced by offering numbers that might highlight the importance of microfluidic devices for containing virus outbreaks. Then, the general reader is presented with ways and commonly used approaches to detect viruses. For a physician/clinical readership, we present many types of signaling alternatives on how these detections are signaled to the operator. Finally, for a technical/engineering reader, fabrication aspects are discussed. We do believe this structure allows for a reading flow independent of readers aim by open this document because it starts scientifically general and becomes more engineering-minded along the text.

6. About the conclusion: I feel the authors should have presented and discussed more scientifically-minded conclusions for various virus detection modes using microfluidics; and then explain how they perceive some of the potential possibilities for COVID19 detection from the existing literature, etc. Indeed, the purpose of this review is to show ways for researchers to exploit microfluidic devices for virus detection and especially in the current scenario of COVID19 detection. This aspect is missing in the current conclusion section

Authors: Thank you for your suggestion. The conclusion is substantially reworked according to those suggestions.

7. General comment: The grammar should be checked throughout.

Authors: Thank you, the authors agreed with checking the grammar.